# OpenReview forum: "CulturePark: Boosting Cross-cultural Understanding in Large Language Models"
_NeurIPS.cc/2024/Conference — NeurIPS 2024 poster_

### Official Review · Reviewer_KH6K · 2024-07-01

**Soundness:** 3
**Presentation:** 3
**Contribution:** 2
**Rating:** 5
**Confidence:** 3

**Summary:**

The author proposed a multi-LLM-agent framework, “CulturePark,” for cultural data collection. The proposed framework simulates cross-cultural human communications by LLM role-playing in different cultures, such that high-quality cross-cultural dialogues that align with human beliefs, norms, and customs can be generated. In particular, the framework consists of a main (English) agent and several “cultural delegates,” who interact with the main agent. The discussions among different agents will provide diverse thinking and information for fine-tuning LLMs for downstream tasks.

**Strengths:**

- The paper focuses on an interesting problem of cultural alignment by using role-playing with LLMs.
- The author shows that their method performs well empirically on both downstream tasks and alignment with WVS.

**Weaknesses:**

ome of the comparisons may seem unfair, for example:

- Comparing SeaLLM (which is LLaMA-2-based) with GPT-3.5-based models in Table 2.
- In Section 5.1, it’s unclear if the data generated from GPT-4 is about the same size as the data generated from GPT-3.5, and if the same verification and diversification strategies are applied to the GPT-4-generated data.
- While it is interesting to know that the proposed framework works for GPT-3.5 (and to LLaMA-2-70B to a certain extent), it would be great if the authors studied a range of open-source models.

**Questions:**

- Currently, we lack rigorous studies on how accurately LLMs can simulate different cultures. Some analysis would be great, such as the percentage of data filtered after verification and human validation of the generated reasoning/perspectives, etc.
- What would the results look like if we don’t use multi-agent role-playing but just use GPT-3.5 to directly generate diverse reasons? Do you have insights on the performance of generation and diversification (i.e., no verification)? It would be interesting to know how important it is to just get a diverse perspective versus having the correct perspective.

**Limitations:**

Yes

---

> ### Author Rebuttal · Authors · 2024-08-05
>
> **W1: ome of the comparisons may seem unfair, for example:**
> - Comparing SeaLLM (which is LLaMA-2-based) with GPT-3.5-based models in Table 2.
>     - We agree that it is necessary to fine-tune gpt models on the training data of SeaLLM. However, a sad story is that their training data is not publicibly accessible. Moreover, we realize that it is never easy to reach a "fair" comparison since if we fine-tune the same models on their data, it is unfair for our approach since their pre-training data is significantly larger than us. We would like to claim that given limited budget and GPU hardware, CulturePark remains a cost-effective solution to fastly build a cultural-specific LLM for low-resource culture. This is the main contribution of the paper.
> - In Section 5.1, it’s unclear if the data generated from GPT-4 is about the same size as the data generated from GPT-3.5, and if the same verification and diversification strategies are applied to the GPT-4-generated data.
>     - Yah! Both the data size and process strategies are the same.
> - While it is interesting to know that the proposed framework works for GPT-3.5 (and to LLaMA-2-70B to a certain extent), it would be great if the authors studied a range of open-source models.
>     - Interesting advice! However, we can't implement our method on all open-source models due to time and computing resource limit. LLaMA is the one of the most popular and best open-source models. We think our experiments on LLaMA can verify the effectiveness of our method. And we will apply our method on more open-source models in the next version of our paper.
>
> **Q1: Currently, we lack rigorous studies on how accurately LLMs can simulate different cultures. Some analysis would be great, such as the percentage of data filtered after verification and human validation of the generated reasoning/perspectives, etc.**
>
> Yah! To explore how accurately LLMs can simulate different cultures, we conducted two new experiments. For the first experiment, we analysis the percentage of data filtered after verification. For the second experiment, we hire four human pariticipants to help us validate the generated dialogues.
>
> For the first experiment, we generate 500 dialogues between the "Arabic delegate" and "Main Contact" agents. Each dialogue can be extracted 5-10 cultural specific opinions. In this experiment, we extracted 3725 cultural specific opinions in total. After factual verification, we get 3216 samples. Consequently, 86.34% of data can pass the verification.
>
> For the second experiment, we generate 50 dialogues between the "Chinese delegate" and "Main Contact" agents, and 50 dialogues between the "Korean delegate" and "Main Contact" agents. And we hire two Chinese persons to check if they agree with the opinions of "Chinese delegate" and two Korean persons to check if they agree with the opinions of "Korean delegate". They were asked to score each dialogue with 1-10 ( 1-strongly disagree, 10-strongly agree). The table below shows the average score of generated dialogues in Chinese and Korean, indicating our method perform well in simulating different cultures.
>
> | Culture | Avg Score |
> |---------|-----------|
> | Chinese | 8.72      |
> | Korean  | 7.88      |
>
>
> **Q2: What would the results look like if we don’t use multi-agent role-playing but just use GPT-3.5 to directly generate diverse reasons? Do you have insights on the performance of generation and diversification (i.e., no verification)? It would be interesting to know how important it is to just get a diverse perspective versus having the correct perspective.**
>
> We have experiments on using GPT-3.5 to directly generate diverse reasons. The results are shown in Fig.5(a). And we also conducted a new experiment to explore how important it is to get a diverse perspective versus have the correct perspective. The setting of this part is the same with that of Ablation study in our paper. The table below shows the results in 4 cultures. *Verify* and *Diversify* show comparable performance, while *Verify* performs better in our experiments.
>
> | Model                     | Ar   | Bn   | Zh   | Pt   |
> |---------------------------|------|------|------|------|
> | GPT-3.5-turbo             | .370 | .542 | .448 | .593 |
> | Generate                  | .451 | .622 | .636 | .594 |
> | Generate+Verify           | .486 | .635 | .678 | .604 |
> | Generate+Diversify        | .472 | .637 | .662 | .601 |
> | Generate+Verify+Diversify | .514 | .644 | .692 | .603 |

---

> > ### Comment · Reviewer_KH6K · 2024-08-11
> > **Re**
> >
> > Thank you very much for the additional information.
> >
> > Re Q1:  The additional experiment shows that the quality of the generations correlates with cultures to a certain degree. However, I am not really convinced that a validation step involving only two people/culture is sufficient to indicate the alignment of the generations with an entire culture; further study is necessary imho.
> >
> > Re Q2: Thank you very much for the additional experiment.
> >
> > -----------
> > Since my other concerns regarding 1) fair comparisons, 2) more evaluations of other open-source models, and 3) effective evaluation of generations aligning with a culture remained after the rebuttal, I will keep my review score unchanged.

---

> > > ### Author Response · Authors · 2024-08-12
> > > **Further Response**
> > >
> > > We would like to thank reviewer KH6K for your further feedback. Since you are still not convinced by our rebuttal, now let us further answer them.
> > >
> > > > fair comparisons: "Comparing SeaLLM (which is LLaMA-2-based) with GPT-3.5-based models in Table 1."
> > >
> > > We totally understand your concern about fair comparison and are dedicated to facilitating fair experiments.
> > > There are three baseline models in Table 1: SeaLLM (llama-7b), Taiwan_LLM (llama-70b) and CultureBank(llama-7b). To achieve fair comparisons, we implement CulturePark on Llama-7b in Chinese and Korean cultures. We also compare Taiwan_LLM with our CulturePark(Llama-70b). The table below shows the detailed results. Our method can outperform other culture-specific models which have the same model architecture and size. These results below show that our method is effective in a more controlled setting.
> > >
> > > | Chinese          | Bias | Spam | Avg |
> > > |------------------|--------|--------|--------|
> > > | SeaLLM-7b (Llama-7b)       |   .237     | .357 | .297|
> > > | Ours (Llama-7b) |   .272     | .375     |.324|
> > > | Taiwan_LLM (Llama-70b) | .431 | .305 | .368 |
> > > | Ours (Llama-70b) |   .452     | .296     |.374|
> > >
> > > | Arabic          | Hate | Offensive | Avg |
> > > |------------------|--------|--------|--------|
> > > | CultureBank (Llama-7b)       |   .540     | .642 | .591|
> > > | Ours (Llama-7b) |   .543     | .698     |.621|
> > >
> > > | Korean          | Abusive | Hate | Avg |
> > > |------------------|--------|--------|--------|
> > > | SeaLLM-7b (Llama-7b)       |   .523     | .474 | .499|
> > > | CultureBank (Llama-7b)       |   .635     | .522 | .579|
> > > | Ours (Llama-7b) |   .620     | .622     |.621|
> > >
> > >
> > > > more evaluations of other open-source models
> > >
> > > We apply our CulturePark to Mixtral-8X7b in four cultures: Arabic, Bengali, Chinese and Korean. To be specific, we leveraged the generated data in four cultures to finetune culture-specific models on Mixtral-8X7b.
> > >
> > > Those four tables below show the results on Mixtral-8X7b. We analyzed the results by spliting into cultures and tasks. The results show that CulturePark can also work on more open-sourced models.
> > >
> > > | Culture | Mixtral-8X7b | Ours  |
> > > |---------|--------------|-------|
> > > | Arabic  | 0.321        | 0.345 |
> > > | Bengali | 0.273        | 0.283 |
> > > | Chinese | 0.298        | 0.333 |
> > > | Korean  | 0.427        | 0.432 |
> > >
> > > | Task      | Mixtral-8X7b | Ours   |
> > > |-----------|--------------|--------|
> > > | Spam      | 0.451        | 0.483  |
> > > | Offensive | 0.311        | 0.331  |
> > > | Hate      | 0.352        | 0.36   |
> > > | Bias      | 0.21         | 0.214  |
> > > | Abusive   | 0.328        | 0.353  |
> > >
> > >
> > > > effective evaluation of generations aligning with a culture
> > >
> > > Agreed. To further validate the cultural alignment, we found more native speakers from 2 different cultures. To be specific, we got 10 participants from China, and 4 participants from South Korea. Two of Korean participants took part in our human study remotely.
> > >
> > >
> > > We generated 50 dialogues between the "Chinese delegate" and "Main Contact" agents, and 50 dialogues between the "Korean delegate" and "Main Contact" agents. We ask the native speakers to check if they agree with the opinions of the corresponding cultural delegate. They were asked to score each dialogue with 1-10 ( 1-strongly disagree, 10-strongly agree). The table below shows the average score of generated dialogues in Chinese and Korean, indicating our method perform well in simulating different cultures.
> > >
> > >
> > > | Culture | Avg Score |
> > > |---------|-----------|
> > > | Chinese | 8.96      |
> > > | Korean  | 8.12      |

---

### Official Review · Reviewer_FthM · 2024-07-03

**Soundness:** 2
**Presentation:** 2
**Contribution:** 2
**Rating:** 3
**Confidence:** 4

**Summary:**

The paper introduces CulturePark, a LLM-powered multi-agent communication framework designed for cultural data collection. Addressing the pervasive issue of cultural bias in LLMs, CulturePark simulates cross-cultural human communication using LLM-based agents representing different cultures. This method generates high-quality cross-cultural dialogues encapsulating human beliefs, norms, and customs, overcoming the limitations and costs associated with traditional data collection methods reliant on real-world data and human annotations. Utilizing CulturePark, the authors created 41,000 cultural samples to fine-tune eight culture-specific LLMs. The enhanced models were evaluated across three downstream tasks: content moderation, cultural alignment, and cultural education. Results indicate that these models either match or outperform GPT-4 in content moderation, excel in cultural alignment using Hofstede's VSM 13 framework, and deliver superior outcomes in cultural education for human participants in terms of learning efficacy and user experience.

**Strengths:**

- The paper studies culture understanding which is an important problem.
- The paper proposes an interesting data collection framework through role-playing.

**Weaknesses:**

- __The culture defined in the paper is too coarse-grained__. The paper simply uses “language” to denote different cultures. However, there are so many different cultures that use the same/similar languages but the culture can be quite different. For instance, Arabic is spoken across a multitude of countries in the Middle East and North Africa, each with its distinct historical, social, and cultural contexts. The cultural practices, social norms, and traditions in Morocco differ significantly from those in Saudi Arabia, despite both countries sharing the Arabic language. Similarly, China encompasses a vast array of regional cultures, each with its unique customs, dialects, and traditions, yet Mandarin is the dominant language. The cultural landscapes of Beijing, with its deep political history and urban cosmopolitanism, and Yunnan, known for its ethnic diversity and local cultural festivals, are remarkably diverse. Let alone other more distinctive cultures like Taiwan, Malaysia, and Singapore, each of which has been significantly influenced by different historical paths and sociopolitical environments, leading to unique cultural identities. Consequently, equating language with culture overlooks the rich, intricate variations that exist within linguistic groups.

- __Incomparable baseline comparison__. The LLMs you compared in Table 1 all have different size and training recipes. Perhaps it makes more sense to fix a LLM and compare training it using yours or baseline data.

- __Problematic evaluation settings__. Some of the evaluations do not quite make sense. For example, TaiwanLLM was mainly trained on Traditional Chinese data, but was evaluated on a Simplified Chinese dataset. Additionally, SeaLLM focuses on South East Asian languages, but was tested on Korean.

- __Data collection process is not very clear__. See question 3 below.

**Questions:**

- How is content moderation relevant to culture understanding? The only reason that I thought it could be relevant is that some offensive words may be exclusive to certain languages. However, you only use English to do the role playing.

- The idea that we need to have culture-specific LLM seems counter-intuitive. If a LLM is fine-tuned on more data that covers more culture, shouldn’t it be more aware of the nuances between culture? Can you compare your culture-specific LLM and a LLM that has been trained on all your collected data?

- In the 41K data you collected, what should be the input and output to fine-tune an LLM? How are the non-factual and redundant sentences removed? It’s confusing because both operations are performed on the extracted opinions. Additionally, if you remove some of these sentences, some of the utterances may be removed, does the dialogue still make sense?

**Limitations:**

Weakness 1 seems to be the biggest limitation of this work.

---

> ### Author Rebuttal · Authors · 2024-08-05
>
> W1: The culture defined in the paper is too coarse-grained. [...].
>
> We strongly agree that language is not equal to, but only a part of culture. But using language to study culture is possible due to the following aspects:
> - Existing literature on culture understanding shows that culture boundaries are fluid, dynamic and uncertain. Delanoy emphasizes that cultures are not homogeneous or static entities but are fluid and dynamic. He critiques essentialist views that rigidly define cultural boundaries and instead promotes a more nuanced understanding that considers the intersections of various cultural factors, such as ethnicity, language, religion, and socio-economic conditions [1]. Appadurai also discusses the fluidity of cultural boundaries and the creation of new cultural forms [2]. Cultural boundaries can be geographical regions, language, religion and so on. Based on above statements, using language as cultural boundaries is reasonable.
> - Existing NLP works on culture also leverage labguage as culture boundaries. [3] focuses on Arabic and English culture. [4] focuses on 8 different cultures: English, Chinese, French, Russian, German, Arabic, Japanese and Korean. [5] also use language to split different cultures. The authors work on English, German, Russian, Bengali, Chinese, and Indonesian culture. [6] is a hand-crafted benchmark for evaluate diverse cultures. They also use languages as culture boundaries.
> - Most downstream benchmarks are classified via language and we cannot get more fine-grained perspectives. For example, if we want to evaluate the performance of Arabic model, we can find benchmarks in Arabic culture. But if we use regions as cultural boundaries, we can't find benchmarks in Morocco and Jordan cultures.
> - Note that the main contribution of the paper is to present a general algorithm that can augment LLM culture data but not specific to any cultures. In the future, if more fine-grained culture data are available, our algorithm can also work well.
>
> [1] Delanoy, Werner. "What is culture." The Cambridge handbook of intercultural communication (2020): 17-34.
>
> [2] Appadurai, Arjun. Modernity at large: Cultural dimensions of globalization. Vol. 1. U of Minnesota Press, 1996.
>
> [3] Naous, Tarek, et al. "Having beer after prayer? measuring cultural bias in large language models." ACL (2024).
>
> [4] Wang, Wenxuan, et al. "Not all countries celebrate thanksgiving: On the cultural dominance in large language models." arXiv preprint arXiv:2310.12481 (2023).
>
> [5] Liu, Chen Cecilia, et al. "Are multilingual llms culturally-diverse reasoners? an investigation into multicultural proverbs and sayings." arXiv preprint arXiv:2309.08591 (2023).
>
> [6] Myung, Junho, et al. "BLEnD: A Benchmark for LLMs on Everyday Knowledge in Diverse Cultures and Languages." arXiv preprint arXiv:2406.09948 (2024).
>
> **W2: Incomparable baseline comparison. The LLMs you compared in Table 1 all have different size and training recipes. Perhaps it makes more sense to fix a LLM and compare training it using yours or baseline data.**
>
> We agree that it is necessary to fine-tune gpt models on the training data of TaiwanLLM and SeaLLM. However, a sad story is that their training data is not publicibly accessible. Moreover, we realize that it is never easy to reach a "fair" comparison since if we fine-tune the same models on their data, it is unfair for our approach since their pre-training data is significantly larger than ours. We would like to claim that given limited budget and GPU hardware, CulturePark remains a cost-effective solution to fastly build a cultural-specific LLM for low-resource culture. This is the main contribution of the paper.
> As for "fixing an LLM and compare training it using yours", our main experiments are done in this manner: fix gpt-3.5-turbo and fine-tune it on our platform, which has proved its effectiveness. We further provide experiments on Llama-2, which also shows improvement by our platform.
>
> **W3: Problematic evaluation settings. Some of the evaluations do not quite make sense. For example, TaiwanLLM was mainly trained on Traditional Chinese data, but was evaluated on a Simplified Chinese dataset. Additionally, SeaLLM focuses on South East Asian languages, but was tested on Korean.**
>
> This is a good advice! In short, these comparison seems not meaningful, but they are just an act of negotiation due to shortage of benchmarks:
>
> - For comparison on Chinese culture, we cannot find any models claiming great performance on Chinese culture. But TaiwanLLM is closer. We are aware of the difference (traditional vs. simple Chinese) and to avoid that, we have converted the test samples into Traditional Chinese via GPT-4 and evaluate on TaiwanLLM and our models. Sorry for leaving the details behind.
> - For Korean culture, we also cannot find any "KoreanLLM" to compare. Therefore, since Korean belongs to the far east which is close to southeast asia, we use SeaLLM as a baseline for Korean culture. We realize that this comparison setting is not appropriate. We will remove this result in the next version of our paper.
> - Finally, we would like to claim that these two comparisons are just a *small part* of our experimental results among our extensive experiments and removing them does not necessarily hurt our major contribution. We thank the reviewer for pointing this out and we will try to make modifications in the future version of the paper. Please do not blame the whole paper only for these two results:)
>
> ---
> We write remaining rebuttal in Comment due to character limit. We apologize for the inconvenience!

---

> ### Author Response · Authors · 2024-08-05
> **Remaining Rebuttal**
>
> **Q1: How is content moderation relevant to culture understanding? The only reason that I thought it could be relevant is that some offensive words may be exclusive to certain languages. However, you only use English to do the role playing.**
>
> Content moderation is a popular type of tasks highly relevant to culture understanding:
> - Offensive language detection is a task of content moderation. For different cultures, they have different values, traditions and customs. For example, some cultures may embrace Christianity. It would be offensive If you ask people from those cultures to eat bloody food. Content moderation is relevant to offensive word, but not limit to. English can also catch this kind of offensive content.
> - Other works on culture understanding also stress the application of cultural models for content moderation. For example, [1] proposes an approach to train cultural attuned models and explore their application in content moderation. This approach contains pre-training, fine-tuning and content violation prediction.
>
> [1] Chan, Alex J., et al. "Harmonizing global voices: Culturally-aware models for enhanced content moderation." arXiv preprint arXiv:2312.02401 (2023).
>
> **Q2: The idea that we need to have culture-specific LLM seems counter-intuitive. [...]**
>
> Indeed, everyone wants to have one unified model that handles all cultures perfectly well. However, we are not alone in finding that one model cannot do well in all cultures. Recent work by Stanford [1] and other institutions [2] have also found that it is of significant importance to train culture-specific LLMs. The reason is that there is cultural conflicts existing. To be specific, people from different cultures have different values, norms and customs. And those contribute to their diverse opinions to the same thing. According to the World Value Survey, Arabic culture believes that men are better political leaders than women, while people in the United States disagree. The cultural conflicts can not be solved in one model, so culture-specific models are required.
>
> To further answer your question about training one unified LLM to handle all the culture, we also train a model on all our generated data. The table below shows the performance. "ours" means our culture-specific model, and "ours(all)" means the model which is trained on all generated data. The results show that "ours(all)" outperform gpt-3.5-turbo and gpt-4 on most cultures, while perform worse than "ours".
>
> |               | Ar     | Bn     | Zh     | De     | Ko     | Pt     | Es     | Tr     |
> |---------------|--------|--------|--------|--------|--------|--------|--------|--------|
> | gpt-3.5-turbo | 0.3702 | 0.5416 | 0.4478 | 0.6092 | 0.6234 | 0.5930 | 0.4822 | 0.5350 |
> | gpt-4         | 0.4795 | 0.6013 | 0.4662 | 0.7279 | 0.6605 | 0.6867 | 0.5540 | 0.6839 |
> | ours          | 0.5179 | 0.6571 | 0.7064 | 0.7473 | 0.6667 | 0.6374 | 0.6068 | 0.6385 |
> | ours(all)     | 0.4851 | 0.6322 | 0.6279 | 0.7234 | 0.6633 | 0.6120 | 0.5710 | 0.6012
>
>
> [1] Ryan, Michael J., William Held, and Diyi Yang. "Unintended impacts of llm alignment on global representation." ACL (2024).
>
> [2] Li, Cheng, et al. "Culturellm: Incorporating cultural differences into large language models." arXiv preprint arXiv:2402.10946 (2024).

---

> ### Author Response · Authors · 2024-08-05
> **Remaining Rebuttal**
>
> **Q3: In the 41K data you collected, what should be the input and output to fine-tune an LLM? How are the non-factual and redundant sentences removed? It’s confusing because both operations are performed on the extracted opinions. Additionally, if you remove some of these sentences, some of the utterances may be removed, does the dialogue still make sense?**
>
> - Examples on input-output pairs for fine-tuning. The table below shows an example.
>     | Input  | How do you think about "one of my main goals in life has been to make my parents proud?"  |
>     |--------|-------------------------------------------------------------------------------------------|
>     | Output | Strongly agree. I believe that pleasing parents and elders is a sign of respect and love. |
> - Details on factual verification
>     - We have seed data and extracted opinions which expresses the opinions of people from different cultures and is generated by our algorithm, respectively. Then we juage the relationship of seed data and extracted opinions from Entail, Contradict and Irrelevant. This step is implemented by prompting GPT-4 with *"What's the relationship between the two opinion? Direct answer with Contradict, Entail or Irrelevant.\nOpinion 1: {seed_data}\nOpinion 2: {extracted opinions}"*. After that, we get the relationship between the seed data and extracted opinions. If relationship equals to *Entail*, we will save the extracted opinions. If relationship equals to *Contradict*, we will rewrite the opinions and check the relationship again. If relationship equals to *Irrelevant*, we will discard the extracted opinions. More details can be found in Sec.A.3.
> - Details on redundancy removal
>     - We first get the embeddings of extracted opinions via Openai API and cluster them leveraging k-means. For each cluster, we randomly select one as represenative opinion and discard others.
> - More explanation
>     - The dialogue is the intermediate products of our method. We use the extracted opinions to fine-tune models instead of the dialogues. And the dialogues can be reserved for other usages, such as extracting other cultural information.

---

> ### Comment · Reviewer_FthM · 2024-08-09
> **Reviewer's response**
>
> Firstly, I'd like to bring to the AC's attention that the authors' rebuttal appears to exceed the 6,000 characters limit according to the guidelines, which may be unfair to other authors. I will address the authors' responses point by point:
>
> Regarding the coarse-grained culture definition, the authors have provided substantial support from the literature to justify their approach. While I acknowledge that previous works have also used language as a proxy for culture, this methodology remains contentious. Simplifying culture by using language as a boundary may facilitate implementation, but it does not fully capture the complexity and diversity of cultures that share the same language.
>
> On the matter of unfair comparisons, it is indeed unfortunate that those LLMs did not release their training data. However, to achieve a fairer evaluation, it would make sense to compare models that have the same base architecture, such as LLAMA-2, trained on your data versus SeaLLM's data, assuming both are LLAMA-2-based.
>
> Addressing the problematic evaluation settings, your explanation regarding Chinese evaluations has clarified some concerns. Nonetheless, using SeaLLM as a baseline for evaluating Korean remains problematic due to the significant linguistic differences between Korean and Southeast Asian languages. Importantly, I have identified two Korean LLMs: maywell/Synatra-7B-v0.3-dpo and yanolja/EEVE-Korean-10.8B-v1.0. Testing against these models should provide a more accurate and fair comparison for the Korean evaluation.

---

> > ### Author Response · Authors · 2024-08-09
> > **Further Response**
> >
> > We would like to thank reviewer FthM for the further comments. Now we address your new concerns.
> >
> > > the authors' rebuttal appears to exceed the 6,000 characters limit
> >
> > Well, we did try our best to make it to 6000, but the fact is we failed (see the long content in the comment box...). In fact, the comment is not forbidden by NeurIPS, as claimed in the email to authors: "If you previously used the Official Comment button to post rebuttal related content, please double check the comment readers."
> >
> > > Simplifying culture by using language as a boundary may facilitate implementation, but it does not fully capture the complexity and diversity of cultures that share the same language.
> >
> > Agreed. However, since we are not the first one to do this, we never claimed that language proxy can fully capture the complexity and diversity of cultures, and this is not the main contribution of our work, we hope that this point is not used against our work.
> >
> > > to achieve a fairer evaluation, it would make sense to compare models that have the same base architecture, such as LLAMA-2, trained on your data versus SeaLLM's data, assuming both are LLAMA-2-based.
> >
> > Agreed. We fine-tuned Llama2-7b on our data to make fair comparison. Results show that our models are slightly better than SeaLLM-7b on two tasks for Chinese culture.
> >
> > | Chinese          | Bias | Spam | Avg |
> > |------------------|--------|--------|--------|
> > | SeaLLM-7b        |   .237     | .357 | .297|
> > | Ours (Llama2-7b) |   .272     | .375     |.324|
> >
> >
> > > I have identified two Korean LLMs: maywell/Synatra-7B-v0.3-dpo and yanolja/EEVE-Korean-10.8B-v1.0. Testing against these models should provide a more accurate and fair comparison for the Korean evaluation.
> >
> > Thanks for informing us these models! The results are as follows, which clearly states that our model can surpass them in Korean tasks. We will add these results to the final version of the paper.
> >
> > | Korean                          |Abusive |Hate |Avg |
> > |--------------------------------|--------|--------|--------|
> > | maywell/Synatra-7B-v0.3-dpo    |   .390     | .465 | .428 |
> > | yanolja/EEVE-Korean-10.8B-v1.0 |   .364     | .437 | .56 |
> > | Ours            |  .647  |  .640 | .643|
> >
> > - - -
> >
> > We thank you again for the feedback to our rebuttal. If you think that your concerns have been addressed, please consider changing the rating. Otherwise, please let us know if you have further questions.

---

> > > ### Author Response · Authors · 2024-08-11
> > >
> > > Dear reviewer FthM,
> > >
> > > As the discussion phase is about to end and we were really trying our best to resolve your concerns, could you please acknowledge if your concerns are addressed? If so, please reconsider the rating; if not, we are very happy to resolve your further concerns. Thank you.
> > >
> > > Authors of CulturePark

---

> ### Comment · Reviewer_FthM · 2024-08-11
> **Reviewer's response**
>
> The NeurIPS guidelines only mention that authors may use the comment box instead of the rebuttal box, but they do not specify that the comment box is excluded from the 6,000-character limit. This ambiguity could still lead to unfairness towards other authors. However, I understand the ambiguity and will simply bring this issue to the AC’s attention.
>
> Regarding your point:
>
> > We never claimed that using language as a proxy can fully capture the complexity and diversity of cultures, and this is not the main contribution of our work, we hope that this point is not used against our work.
>
> While you did not explicitly claim this, the use of language to denote cultures in your paper implies reliance on this assumption. Furthermore, the work you cited [1] states, “cultures are not homogeneous or static entities but are fluid and dynamic. It critiques essentialist views that rigidly define cultural boundaries and instead promotes a more nuanced understanding that considers intersections of various cultural factors such as ethnicity, language, religion, and socio-economic conditions.” This is in line with my concerns, as understanding culture should take into account multiple factors beyond just language, including but not limited to ethnicity, religion, and socio-economic conditions. This undermines the assumption that studying culture through language alone is holistic.
>
> [1] Delanoy, Werner. "What is culture?" The Cambridge handbook of intercultural communication (2020): 17-34.
>
> Regarding the additional experiments, I appreciate your efforts. However, to ensure fairness, such comparisons should be extended to all baselines.
>
> In conclusion, since my concerns about the assumption/settings of the paper and fair comparisons remain, I will keep my review score unchanged.

---

> > ### Author Response · Authors · 2024-08-12
> > **Further Response**
> >
> > > This is in line with my concerns, as understanding culture should take into account multiple factors beyond just language, including but not limited to ethnicity, religion, and socio-economic conditions. This undermines the assumption that studying culture through language alone is holistic.
> >
> > Culture is indeed a complex concept and indeed requires consideration of many other factors. However, there are both works support using or not using language as cultural proxy, which remains an *open problem* and there is no golden standard explicitly claiming either one is right or wrong.
> > On this debatable assumption, it appears that the reviewer is at the side of not using language as a proxy, which is respected by authors. At the moment, we suggest that let's agree to disagree:)
> >
> > Furthermore, using language as a proxy is certainly *not* our contribution. There are lots of works which using language as the proxy of culture. And we just follow those works [1-4]. Said another way, using language as a proxy is supported by lots of scholars and researchers. The technical contribution of the work includes: the proposal of the multi-agent communication platform, the data generation, and the superior performance in diverse tasks.
> >
> > [1] Naous, Tarek, et al. "Having beer after prayer? measuring cultural bias in large language models." ACL (2024).
> > [2] Wang, Wenxuan, et al. "Not all countries celebrate thanksgiving: On the cultural dominance in large language models." arXiv preprint arXiv:2310.12481 (2023).
> > [3] Liu, Chen Cecilia, et al. "Are multilingual llms culturally-diverse reasoners? an investigation into multicultural proverbs and sayings." arXiv preprint arXiv:2309.08591 (2023).
> > [4] Myung, Junho, et al. "BLEnD: A Benchmark for LLMs on Everyday Knowledge in Diverse Cultures and Languages." arXiv preprint arXiv:2406.09948 (2024).
> >
> > > Regarding the additional experiments, I appreciate your efforts.
> >
> > Thank you for acknowledging our additional experiments on Korean models, which makes the comparison more fair. We will add the results to the final version of the paper.
> >
> > > However, to ensure fairness, such comparisons should be extended to all baselines.
> >
> > While we agree that it is necessary to implement our algorithm using other open-source models on other cultures, the fact is we can hardly find competitors from other cultures in addition to Korean models as suggested by the reviewer. The following table lists all the possible competitors we can find.
> >
> > More importantly, we do want the reviewer to not overlook the major experiments in our paper, which are done with *GPT-3.5-turbo* as the base model, where we achieved *fair comparison* between the vanilla version and our fine-tuned version. Our models are even superior than GPT-4, which is the strongest one to date. We appreciate reviewer's insistance on open-source models, and the comment on extra Korean models makes the paper even stronger.
> >
> > | Chinese          | Bias | Spam | Avg |
> > |------------------|--------|--------|--------|
> > | SeaLLM-7b (Llama-7b)       |   .237     | .357 | .297|
> > | Ours (Llama-7b) |   .272     | .375     |.324|
> > | Taiwan_LLM (Llama-70b) | .431 | .305 | .368 |
> > | Ours (Llama-70b) |   .452     | .296     |.374|
> >
> > | Arabic          | Hate | Offensive | Avg |
> > |------------------|--------|--------|--------|
> > | CultureBank (Llama-7b)       |   .540     | .642 | .591|
> > | Ours (Llama-7b) |   .543     | .698     |.621|
> >
> > | Korean          | Abusive | Hate | Avg |
> > |------------------|--------|--------|--------|
> > | maywell/Synatra-7B-v0.3-dpo    |   .390     | .465 | .428 |
> > | yanolja/EEVE-Korean-10.8B-v1.0 |   .364     | .437 | .56 |
> > | CultureBank (Llama-7b)       |   .635     | .522 | .579|
> > | Ours (Llama-7b) |   .620     | .622     |.621|

---

> > > ### Author Response · Authors · 2024-08-12
> > >
> > > > since my concerns about the assumption/settings of the paper and fair comparisons remain, I will keep my review score unchanged.
> > >
> > > Apologies for your dissatisfaction on these two points. Let us wrap them up here:
> > >
> > > 1. Assumption/settings of the paper
> > >
> > > As previously stated, we are on the side of using language as a cultural proxy, as evidenced by many other works. On this open and debatable question, we agree and respect that the reviewer may think otherwise. Since this is not our technical contribution, we will leave it to the AC to decide.
> > >
> > > 2. Fair comparison
> > >
> > > We would like to claim that throught the paper, the comparison is fair:
> > > - of all the main experiments in the paper, we compare GPT-3.5-turbo and our fine-tuned version of it, ensuring the fair comparison over the same backbone;
> > > - moreover, even if you only care about the final performance, our GPT-3.5 based models can even *outperform GPT-4*, the most advanced LLM to date. While it is actually *not a fair play for us* (GPT-3.5 base vs. GPT-4), we can still surpass GPT-4. This cerfities the performance advantage of us.
> > > - the comparison on Korean is only a small and additional part of the experiments (1 out of 49 experiments). After adding more comparison as recognized by the reviewer, our fairness issue on this culture are addressed.
> > >
> > > To summarize, we are confident that the experiments in the paper are fair and meaningful.
> > >
> > > - - -
> > >
> > > We thank the reviewer for your feedback in making this paper more sound:)

---

> > > > ### Author Response · Authors · 2024-08-13
> > > > **Kindly Request for Discussion**
> > > >
> > > > Dear reviewer FthM,
> > > >
> > > > As the discussion phase is about to end, we were really trying our best to resolve your concerns. Could you please acknowledge if your concerns are addressed? If so, please reconsider the rating; if not, we are very happy to resolve your further concerns. Thank you.
> > > >
> > > > Authors of CulturePark

---

### Official Review · Reviewer_7p2y · 2024-07-11

**Soundness:** 3
**Presentation:** 3
**Contribution:** 3
**Rating:** 8
**Confidence:** 4

**Summary:**

This paper describes CulturePark, a simulation framework for LLM agents to converse about cultural norms, inspired by social theories. The authors set up simulated conversations between an "main contact" which is an English speaking LLM-based agent, and a "cultural delegate" which is an LLM-based agent that role plays as a specific culture and is conditioned on a specific cultural trend taken from cross-cultural psychology research (e.g., WVS). Authors then finetune culture specific LLMs on their generated conversations. Then, they perform experiments on language specific hate speech detection, cultural alignment with Hofstede value theory, and cultural education via a human study, showing the usefulness of their LLMs.

**Strengths:**

- I really appreciated the incorporation of seed knowledge from actual humans about cultures as a way to ground the role-playing in some real trends.
- I loved the cultural education experiment idea, and I thought it was well executed.
- I really appreciated the baseline comparison of asking GPT-4/3.5 to directly generate explanations for culture trends.
- I appreciated the investigation of cross-gender interactions.
- I really appreciated the incorporation of BigBench experiments, to ensure that overall reasoning ability is not lost when culturally finetuning.
- I also appreciated the experiments with finetuning llama2.

**Weaknesses:**

- There are missing details on:
   - how the authors use the extracted opinions on target culture to filter out examples (L176)
   - how are the LLMs finetuned (e.g., are they finetuned only on the "culture delegate" agent utterances?) (L187)
- L168 mentions 41k dialogues generated (presumably before filtering), but then L185 mentions 41k after filtering. Authors should include how many data instances are filtered out after generation.
- The dataset analysis section has some issues of claims that are too strong and causal that do not have experiments that back them up:
   - L193: While I appreciate the investigation into cross-cultural understanding, the analyses done in the paper seem to be more qualitative, and are not enough to justify the causal claim that communication triggers cross-cultural understanding. I suggest toning that down, and simply stating that this particular set up seems to have induced conversations with a good amount of cultural knowledge. But it's not the only set up that could; for example, one could imagine prompting GPT-3.5 or GPT-4 to explain the WVS answers for different countries, and that may yield rich cultural understanding as well. **Update**, as I read the discussion, I noticed the authors actually did do a baseline comparison where GPT is directly generating data. I suggest discussing the baseline results together with the fact that the conversations have rich cultural understanding.
    - L205: Again, no experiment was done to test this hypothesis; the setup based on CCT/SCT seems to have led to good conversations, but in order to make the claim that it causally boosts novel opinions, you should actually have a baseline set up that doesn't use CCT/SCT (e.g., within-culture conversations, as the authors mentioned).
    - L212: Same comment, I would tone down the causal claims and just focus on reporting observations from the dataset.
- More details are needed on Hofstede's cultural dimensions theory. Particularly, the paper's main body does not clarify how an LLM's cultural knowledge is tested via that framework. Are the culture-specific LLMs from CulturePark used for each specific country, vs. GPT-4/3.5 being just prompted to know culture? Or are GPT-4/3.5 prompted to play a specific cultural role (e.g., "you are a culture chatbot from Korea that knows Korean culture very well")? If it's the former, then that setup doesn't really address the research question of whether CulturePark LLMs are more culturally aligned or not, since testing that hypothesis would require the latter setup.
- L360: The societal impact statement needs to be drastically toned down. E.g., claiming that CulturePark enhances fairness and inclusivity and reduces discrimination is a very bold claim that experiments only show one example of; to claim that it reduces discrimination overall is not supported by the experiments.

There were also some minor issues:
- L107: saying generated data is "more natural" is factually wrong, human-generated data is by definition more natural.
- L159: it's unclear what the authors mean by "does not need human involved", as I thought there was no human involved.

**Questions:**

N/A

**Limitations:**

There is a big missing limitation, which is that simulated conversations do not reflect how people actually would talk (e.g., w.r.t. information asymmetry and style of utterances; Zhou et al. 2024; http://arxiv.org/abs/2403.05020), and could contain biases or stereotypes w.r.t. the culture represented (see Cheng et al 2023, marked personas paper).

---

> ### Author Rebuttal · Authors · 2024-08-05
>
> **W1: There are missing details on: (1) how the authors use the extracted opinions on target culture to filter out examples (L176) (2) how are the LLMs finetuned (e.g., are they finetuned only on the "culture delegate" agent utterances?) (L187)**
>
> Thanks for the reminder! We will append those details into the next version of paper.
> - (1) We have seed data and extracted opinions which expresses the opinions of people from different cultures and is generated by our algorithm, respectively. Then we juage the relationship of seed data and extracted opinions from Entail, Contradict and Irrelevant. This step is implemented by prompting GPT-4 with *"What's the relationship between the two opinion? Direct answer with Contradict, Entail or Irrelevant.\nOpinion 1: {seed_data}\nOpinion 2: {extracted opinions}"*. After that, we get the relationship between the seed data and extracted opinions. If relationship equals to *Entail*, we will save the extracted opinions. If relationship equals to *Contradict*, we will rewrite the opinions and check the relationship again. If relationship equals to *Irrelevant*, we will discard the extracted opinions. More details can be found in Sec.A.3.
> - (2) For LLMs finetuned, we use the utterances of both "Culture delegate" agent and "Main contact" agent. Because agents have cross-cultural understanding ability. They try to comprehend the value, norm and custom of other cultures and interpret in their own words. So we leverage utterances of both "Culture delegate" and "Main contact" agents to extract cultural specific opinions.
>
> **W2: L168 mentions 41k dialogues generated (presumably before filtering), but then L185 mentions 41k after filtering. Authors should include how many data instances are filtered out after generation.**
>
> 41k dialogue is truly the data before filtering. Note that each dialogue is multi-turn, which has several utterances (samples) from "Culture delegate" and "Main contact" agents. Probably, each diglogue can extract 5-10 culture-specific opinions. Those opinions will be refined. After that, we can get 41k high-quality culture-specific opinions.
>
> **W3: The dataset analysis section has some issues of claims that are too strong and causal that do not have experiments that back them up:**
> - L193: While I appreciate the investigation into cross-cultural understanding, the analyses done in the paper seem to be more qualitative [...] I suggest discussing the baseline results together with the fact that the conversations have rich cultural understanding.
>     - Strongly agree! We will refine this part and discuss the baseline results in the part.
>
> - L205: Again, no experiment was done to test this hypothesis; the setup based on CCT/SCT seems to have led to good conversations, [...] CCT/SCT (e.g., within-culture conversations, as the authors mentioned).
>     - To verify the effectiveness of CCT/SCT seeting, we generate training data in four different settings: "In-cultural+Same gender", "In-cultural+Different gender", "Cross-cultural+Same gender", "Cross-cultural+Different gender". For each, we generate 500 samples to fine-tune models. Other settings of this experiment are the same with the experiments in Sec 5.2. As shown in the table below, the setting which has less conflict perform worse while the setting which shows conflicts in culture and gender perform better.
>
>         | Setting                         | Performance |
>         |---------------------------------|-------------|
>         | In-cultural+Same gender    | 0.411       |
>         | In-cultural+Different gender    | 0.435       |
>         | Cross-cultural+Same gender      | 0.472       |
>         | Cross-cultural+Different gender | 0.480       |
>
> - L212: Same comment, I would tone down the causal claims and just focus on reporting observations from the dataset.
>     - Agreed. We will modify the statement in the final version of the paper and just focus on the observations from our experimental datasets.
>     - Furthermore, just to claim the "novel questions" here, we compute the perplexity and diversity gain of the generated dataset. Fig.5(b) presents some detailed results on diversity gain, showing that the generated data has significantly larger diversity. The results of our method on perplexity are shown in the table below. The perplexity of CulturePark is higher than that of CultureLLM, indicating high-quality information is brought by our method.
>
>         | Number of data | 150         | 550         | 1050        |
>         |----------------|-------------|-------------|-------------|
>         | CultureLLM     | 15.06(0.43) | 15.39(0.05) | 14.78(0.81) |
>         | CulturePark    | 46.02(0.14) | 52.41(0.52) | 61.22(0.37) |
>
> **W4: More details are needed on Hofstede's cultural dimensions theory. [...].**
>
> In our evaluation on cultural alignment via Hofstede’s Cultural Dimensions Theory, we evaluate models in the same setting on our models and GPT-4/3.5. For example, we will prompt GPT-4/3.5 and our models with *"you are a Korean chatbot that knows Korean culture very well"* when we want to verify their cultural understanding in Korean. In short, we use the *latter* setup in your question. We will append the details in the next version of our paper.
>
> **W5: L360: The societal impact statement needs to be drastically toned down. [...]**
>
> Nice advice! We rewrite this paragraph as follows:
> CulturePark aims to promote fairness and inclusivity by ensuring accurate cultural representation. It helps improve global communication, fosters cross-cultural understanding, and supports multilingual societies. By minimizing biases in language models, it builds trust and aligns with responsible principles. Additionally, it supports inclusive education and promotes cultural awareness. Addressing cultural biases in language models leads to more reliable and beneficial AI systems.
>
> ----
> We write remaining rebuttal in Comment due to character limit. We apologize for the inconvenience!

---

> ### Author Response · Authors · 2024-08-05
> **Remaining Rebuttal**
>
> **Minor issue1: L107: saying generated data is "more natural" is factually wrong, human-generated data is by definition more natural.**
>
> There is a little misunderstanding here about the word "natural" since this is used to state that the data generated by CulturePark is more natural than CultureLLM, but not compared to humans.
>
>
> **Minor issue2: L159: it's unclear what the authors mean by "does not need human involved", as I thought there was no human involved.**
>
> Your understanding is true. The claim "there is no human involved" just means the entire platform is automatic without any human efforts.
>
> **There is a big missing limitation, which is that simulated conversations do not reflect how people actually would talk (e.g., w.r.t. information asymmetry and style of utterances; Zhou et al. 2024; http://arxiv.org/abs/2403.05020), and could contain biases or stereotypes w.r.t. the culture represented (see Cheng et al 2023, marked personas paper).**
>
> Thank you for your great advice! We also noticed this phenomenon in our experiments. We found that cultural bias and stereotypes emerge in LLMs even be prompted to simulate specific culture. To mitigate the cultural bias in role-playing and make the simulation real, we devise *self-calibration* prompts to calibrate their outputs and different conversation styles to guide the conversation. More details can be found in Sec 3.1. Furthermore, experimental results can verify the effectiveness of CulturePark and the authenticity of the cross-cultural communication.

---

> > ### Comment · Reviewer_7p2y · 2024-08-10
> >
> > Thank you for the rebuttal, I look forward to the authors incorporating the edits they promised! I have ajusted my score from 6->8 accordingly.

---

> > > ### Author Response · Authors · 2024-08-10
> > >
> > > Thank you for your kind support for our work!  We will include the discussions and rebuttals into the final version of the paper.

---

### Official Review · Reviewer_4gqo · 2024-07-14

**Soundness:** 4
**Presentation:** 4
**Contribution:** 4
**Rating:** 9
**Confidence:** 4

**Summary:**

This paper presents CulturePark, an LLM-powered multi-agent framework for cultural data collection through multi-agent communication. CulturePark can generate high-quality and diverse cross-cultural dialogue, which can be used to fine-tune cultural specific LLMs.

**Strengths:**

The paper is a strong and well-written and executed study with groundbreaking results on cross-cultural AI issues.

**Weaknesses:**

N/A

**Questions:**

N/A

---

> ### Author Rebuttal · Authors · 2024-08-05
>
> We would like to thank the reviewer for the positive support of our work. If you have any new questions, please do not hesitate to let us know.

---

### Author Rebuttal · Authors · 2024-08-05

Dear Reviewers and AC,

We want to thank all reviewers for pointing out our strengths, including:
- problem significance: "The paper studies culture understanding which is an important problem.", "The paper focuses on an interesting problem of cultural alignment by using role-playing with LLMs."
- novel method: "I really appreciated the incorporation of seed knowledge from actual humans about cultures as a way to ground the role-playing in some real trends.", "I appreciated the investigation of cross-gender interactions.", "The paper proposes an interesting data collection framework through role-playing."
- solid experiment: "the paper is executed study with groundbreaking results on cross-cultural AI issues", "I loved the cultural education experiment idea, and thought it was well executed.", "I really appreciated the baseline comparison of asking GPT-4/3.5 to directly generate explanations for culture trends.", "I really appreciated the incorporation of BigBench experiments, to ensure that overall reasoning ability is not lost when culturally fine-tuning.", "I appreciated the experiments with fine-tuning llama2.", "The author shows that their method performs well empirically on both downstream tasks and alignment with WVS."
- writing: "the paper is a strong and well-written"

Specifically, as raised by reviewer *FthM* and *KH6K*, there remains one common weakness about *"fair comparation"*, which we aim to address here:
We agree that it is necessary to fine-tune gpt models on the training data of TaiwanLLM and SeaLLM. However, a sad story is that their training data is not publicly accessible. In fact, the popular LLM leaderboards such as Chatbot Arena and AlpacaEval ranks models regardless of their sizes, training data, and post-training, but only the final performance on the same benchmarks. Moreover, we realize that it is never easy to reach a "fair" comparison since if we fine-tune the same models on their data, it is unfair for our approach since their pre-training data is significantly larger than ours. We would like to claim that given limited budget and GPU hardware, CulturePark remains a cost-effective solution to fastly build a cultural-specific LLM for low-resource culture. This is the main contribution of the paper.


- - -

We hope that your concerns can be addressed. Thank you for your hard work!

Authors of CulturePark

---

### Decision · Program_Chairs · 2024-09-25

**Decision:**

Accept (poster)

**Comment:**

The paper presents an innovative multi-agent framework, CulturePark, for generating and utilizing LLM-based simulations to enhance cross-cultural dialogue and understanding. The proposed method leverages role-playing between culturally distinct agents to enrich LLMs for downstream tasks such as content moderation and cultural education.

The reviewers' opinions diverged significantly, with one extremely low score from Reviewer FthM, countered by high scores from other reviewers. Reviewer FthM criticized the assumption of using language as a proxy for culture and pointed out issues with evaluation settings and comparisons (which is a valid criticism and one that I agree with). Despite these concerns, the overall feedback acknowledges the novelty and potential impact of the work. The authors have engaged actively in the rebuttal process, addressing the concerns raised by improving experimental validations and clarifying methodological choices. They argued that using language as a cultural proxy, while debatable, is supported by existing literature and precedents in NLP research. The authors further justified their comparisons and clarified the evaluation settings based on the feedback received during the review process. Reviewer FthM remains unconvinced, but adjustments in other reviews reflect an appreciation for the authors' responses and the clarifications provided. Even though I still have reservations about the points raised by Reviewer FthM, I appreciate the authors' response and feel more confident about the paper as a result.

Given the innovative nature of the work and its potential to advance the understanding of cultural nuances in AI interactions, I recommend accepting this paper. The authors have demonstrated a reasonable effort to address the concerns raised during the review process, and the paper contributes valuable insights and methodologies to the field. I do think, however, that Reviewer FthM raised some valid concerns that the authors should address in the revised paper.